# Survival of Lung Cancer Patients by Histopathology in Taiwan from 2010 to 2016: A Nationwide Study

**DOI:** 10.3390/jcm11195503

**Published:** 2022-09-20

**Authors:** Hsuan-Chih Tsai, Jing-Yang Huang, Ming-Yu Hsieh, Bing-Yen Wang

**Affiliations:** 1Department of Family Medicine, Taichung Armed Forces General Hospital, Taichung 41148, Taiwan; 2Department of Occupational Medicine, Taichung Veterans General Hospital, Taichung 40705, Taiwan; 3School of Medicine, National Defense Medical Center, Taipei 11490, Taiwan; 4Center for Health Data Science, Chung Shan Medical University Hospital, Taichung 40201, Taiwan; 5Institute of Medicine, Chung Shan Medical University, Taichung 40201, Taiwan; 6Department of Otorhinolaryngology-Head and Neck Surgery, Changhua Christian Hospital, No. 135, Nanxium St., Changhua 50006, Taiwan; 7Department of Post-Baccalaureate Medicine, College of Medicine, National Chung Hsing University, Taichung 40227, Taiwan; 8Division of Thoracic Surgery, Department of Surgery, Changhua Christian Hospital, Changhua 50006, Taiwan; 9School of Medicine, Chung Shan Medical University, Taichung 40201, Taiwan; 10School of Medicine, College of Medicine, Kaohsiung Medical University, Kaohsiung 80708, Taiwan; 11Institute of Genomics and Bioinformatics, National Chung Hsing University, Taichung 40227, Taiwan

**Keywords:** histopathology, lung cancer, mortality, targeted therapies

## Abstract

Objective: Lung cancer poses a tremendous threat to the modern world. According to Taiwan’s Ministry of Health and Welfare, lung cancer took first place in total cancer deaths in 2021. This study investigated the overall lung cancer survival based on histopathology between 2010 and 2016 in Taiwan. Method: Data from 2010 to 2016 was collected from the Taiwan Cancer Registry (TCR). The characteristics and overall survival of 71,334 lung cancer patients were analyzed according to the tumor, node, metastasis (TNM) 7th staging system. Univariate and multivariate analyses were performed to identify differences in 1-year, 3-year, and 5-year survival between different histopathologies of lung cancer. Results: The 1-year overall survival rate increased from 54.07% in 2010 to 66.14% in 2016. The 3-year overall survival rate increased from 26.57% in 2010 to 41.12% in 2016 in all patients. Among the histopathologies of lung cancer, 3-year overall survival of adenocarcinoma patients increased the most and largely contributed to the increased 3-year overall survival of all lung cancer patients. Conclusions: The introduction of target therapy has led to a tremendous increase in overall survival for lung adenocarcinoma patients. However, target therapy differs by histopathology. Choosing the right target therapy and determining the correct histopathology of lung cancer is a pivotal key in increasing the overall survival of patients. Together with immune therapy, the landscape of lung cancer treatments is changing.

## 1. Introduction

Lung cancer casts an enormous shadow on human health. Lung cancer has been the most lethal cancer, accounting for 18.4% and 18% of the global cancer deaths in 2018 and 2020, respectively. Also, lung cancer was the most diagnosed cancer in 2018 (11.6%) and the second-most diagnosed cancer (11.6%) in 2020 [1,2]. In many countries, the age-standardized five-year net survival rate for patients diagnosed with lung cancer was relatively poor in the CONCORD-2 and CONCORD-3 studies [3,4]. In a European study, the five-year net survival rate of lung cancer patients was 10–17% between 1996–2016 [5]. Few patients were diagnosed at an early stage and could have the opportunity to receive surgical resection.

The treatment strategy for lung cancer varies, but it mostly depends on the staging. We have documented lung cancer and prognosis in Taiwan from 2002 to 2008 on the basis of the 6th edition of the tumor, node, and metastasis (TNM) staging system [6]. In 2009, the 7th edition of the TNM classification for lung cancer was released; it was followed by the 8th edition in 2016. Lung cancer has different cell types, including adenocarcinoma, squamous cell carcinoma, adenosquamous cell carcinoma, and large cell carcinoma, which all could be classified as non-small cell lung carcinoma (NSCLC), and small cell lung carcinoma (SCLC). A Taiwanese study investigated lung cancer patients between 2002–2014; it documented that large cell carcinoma and adenocarcinoma patients had better 5-year survival rates of 30.2% and 22%, respectively [7].

In addition, an important determinant of the 3-year survival rate in lung cancer is histopathology. The World Health Organization (WHO) introduced the latest histological classification in 2021, emphasizing several aspects: (1) the definition and classification of small diagnostic samples, and (2) the utilization of genetic testing to precisely diagnose lung cancer and achieve a better therapeutic outcome [8]. Howlader et al., documented a correlation between mortality and histopathology of lung cancer in North America. There was a decrease in mortality from NSCLC, and the incidence of NSCLC in both men and women decreased. Unfortunately, there was no improvement in SCLC survival [9]. Little literature has investigated the disparity between histopathology of lung cancer and mortality. The aim of this study was to reveal the relation between different histological subtypes and mortality in Taiwan.

## 2. Materials and Methods

This study was approved by the Changhua Christian Hospital Institutional Review Board (number: 161222) and was conducted in accordance with the principles stated in the Declaration of Helsinki. The Taiwan Cancer Registry (TCR) was established and introduced for cancer registration in 1979; it is supervised and funded by the Ministry of Health and Welfare (MOHW) [10]. Commissioned by the government, the TCR provides abecedarian training courses and standardizes detailed items of cancer patients [10]. A long-form database was implemented in the TCR in 2002, and the number of cancer registry items eventually increased from 20 (short form) to 114 (long form) in 2011. The long-form database includes cancer-site-specific factors and several items directly linked to patient care (laboratory values, tumor markers, etc.). The TCR is linked to the National Health Insurance of Taiwan, a social insurance that covers most treatment-related costs, including blood sampling, chest x-rays, chest/abdominal computed tomography scans, operative costs, chemotherapy, target therapy, radiotherapy, and immunotherapy.

We retrieved lung cancer patients’ records from the TCR using the International Classification of Disease for Oncology codes C34.0, C34.1, C34.2, C34.8, and C34.9. All data collected from the TCR represented a newly diagnosed cancer case between 2010 and 2016. Follow-up data was used from the date of diagnosis to either the date of death or the censoring date of 31 December 2018. The following clinical variables were included in the study for analysis: age, sex, Charlson Comorbidity Index, cell type, clinical T, clinical N, clinical M, clinical stage, grading, treatment strategy, and survival data. The initial treatment strategy was defined as the therapy administrated within three months of diagnosis. All tumor specimens were graded histologically based on the World Health Organization’s classification of lung cancers and were staged according to the 7th TNM staging system [10].

### Statistical Analysis

The analysis for this study was performed by using SAS 9.4 for Windows (SAS Institute Inc., Cary, NC, USA). Overall survival was defined from tissue diagnosis to the date of death or 31 December 2018. The date of death and its cause were obtained from a Taiwan death certificate database. The overall survival curve was calculated by the Kaplan-Meier method, and the log-rank test was used to determine significant differences. The Charlson Comorbidity Index was used to quantify preexisting comorbidities [11].

## 3. Results

Throughout the study period, the annual number of lung cancer patients increased from 9260 in 2010 to 11,565 in 2016. There were trends of younger age and more tumors of size < 2 cm, and late-stage diagnoses and men were predominant. The prevalence of adenocarcinoma increased by 11 percentage points, and there were gradual decreases in the prevalence of squamous cell carcinoma (LUSC), large cell carcinoma (LCC), and other cell types of lung cancer.

Most lung cancer patients received some kind of treatment such as chemotherapy, surgery, radiotherapy, or target therapy. Notably, the number of lung cancer patients receiving target therapy dramatically increased from 8 (0.1%) in 2010 to 3450 (29.8%) in 2016. The basic clinical demographic data were summarized in Table 1.

The abbreviation of adenocarcinoma (LUAD) and squamous cell carcinoma (LUSC) was used in accordance with TCGA nomenclature The overall survival (OS) statistics were listed in Table 2. The 1-year OS rate for all patients was above 50%. The 3-year OS rate for all patients gradually increased from 26.57% in 2010 to 41.12% in 2016. From 2010 to 2016, the 3-year OS rates for stage I, stage II, and stage III patients, respectively, increased by roughly 10 percentage points. Even in the terminal stage, the 3-year OS rate increased by 5 percentage points.

Kaplan-Meier (K-M) curves were plotted for reviewing the OS. In Figure 1A, 1-year OS and 3-year OS for all patients increased, with a significant increase in 3-year OS. In addition, the stage-related OS in Figure 1B had the same trend of increasing OS in stage I, stage II, and stage III.

Regarding the histopathological aspect, Figure 2A shows the OS survival related to adenocarcinoma; there were slight increases in 1-year and 5-year OS. However, a 10 percentage point increase in 3-year OS of adenocarcinoma patients was observed. Stage-related 3-year OS was drawn in Figure 2B. There was a notable increase for stage I patients, fluctuations for stage II and stage III patients, and a slight increase for stage IV patients.

Regarding LUSC patients, there were no obvious increases in 1-year, 3-year, and 5-year OS; the same was true for stage-based 3-year OS (Figure 2C,D). Concerning adenosquamous (LUADSC) cell carcinoma patients, there were a gradual increase in 3-year OS and conspicuous increases in 3-year OS for stage II and stage III (Figure 2E,F). Turning to large cell cancer patients, there was no substantial change in 3-year OS, but there was a distinguishable increase in 3-year OS for stage I patients (Figure 3A,B). Small cell carcinoma reflected the worst 3-year OS; however, the 3-year OS increased in early-stage patients (Figure 3C,D).

## 4. Discussion

The study shows an outstanding increase in Taiwan’s lung cancer survival from 2010 to 2016; an increasing OS of stage I to stage IV adenocarcinoma patients contributed the most. Unfortunately, there were no obvious increases in OS in other histopathological types of lung cancer. Another finding was that more detection at an early stage had a contribution to 3-year OS, even for histopathologies with low prognoses. Sheng Hu et al., disclose the same trend in OS of adenocarcinoma patients using the US Surveillance, Epidemiology, and End Results (SEER) database (1998–2018), 27.8% of the US population was covered by the SEER database. They discovered a 41.72% increase in the median survival time of all lung cancer patients. Adenocarcinoma patients had a 5% increase in median survival compared with a 3% increase in squamous patients [12] OS of American population statistics in adenocarcinoma and squamous cell carcinoma was consistent with our study, with a 2.71% increase in the SEER database. We revealed no significant differences in the overall survival curve of squamous patients. (Figure 2C).

Likewise, the five-year relative survival ratio in adenocarcinoma increased both in men and women, 10% in men, and 8% in women respectively from the years 1996–2016. Contrary to the finding in U.S. and Asia, squamous cell carcinoma account mostly for the Estonian population [13]. In northern Europe, adenocarcinoma accounted for half of all lung cancers, the study also saw an improvement in relative survival both in adenocarcinoma and squamous cell carcinoma. On the contrary, squamous cell carcinoma had a better prognosis than adenocarcinoma [14]. The survival differs among different regions around the world. Asian and United States share similar results. Even in northern and eastern Europe, there are different results.

Advanced lung cancer predominates across the whole world. In an eastern European study, a slightly increased localized lung cancer was discovered from 2010 to 2016. The proportion of localized cases of adenocarcinoma increased by approximately 2% in an eastern European study. However, there has probably been a shift toward smaller tumors within this rather well-studied category, and more frequent use of chest computed tomography (CT) for other indications provides a certain contribution, resulting in incidental detection of early tumors. From 2010 to 2016, the proportion of T1 tumors among localized cases showed a 20% increase [13]. China researchers also confirmed with the improvement in diagnostic techniques such as radiologic imaging, and CT, more patients are being detected in an early stage of NSCLC during 1998–2005 [15]. We could estimate earlier stages of NSCLC were detected with the advancement of technology.

In our previous study, only 12.5% of lung cancer patients in Taiwan were clinical stage I between 2002 and 2008 [6]. A more current study of ours had a compatible result: an increasing rate of stage I disease from 13.5% in 2010 to 23.3% in 2016. Likewise, we could see the tumor size below 2 cm increased from 8.7% in 2010 to 14.3% in 2016, respectively. Stage 1A patients increased from 8.6% in 2010 to 14% in 2016. Moreover, the 5-year survival rate of lung cancer patients was 15.9% between 2002 and 2008; it gradually increased and was 25% between 2010 and 2016 [6]. Early detection of lung cancer was the most effective way to improve the survival rate in lung cancer patients.

Lung cancer can be either non-small cell lung carcinoma (NSCLC) or small cell lung carcinoma (SCLC), and the treatment options vary. Treatment for NSCLC depends on the stage, histopathology, genetic alterations, and patient’s condition and can include surgery, radiotherapy, chemotherapy, adjuvant platinum-based chemotherapy, immunotherapy, and molecularly targeted therapy, either alone or in combined modality. Surgical resection with curative intent is recommended for medically fit patients with early-stage NSCLC. Molecular target testing and immune inhibition take a great role in tailoring treatment for advanced NSCLC. Sixty percent of Asians have a mutation in the tyrosine kinase domain of the *EGFR* gene [16]. Therefore, specific targeted therapies have become standard therapy for advanced NSCLC and relapse after primary therapy. The mutation sites include the epidermal growth factor receptor (*EGFR*), B-Raf proto-oncogene (*BRAF*), the echinoderm microtubule-associated protein-like 4 (*EML4*)-anaplastic lymphoma kinase (*ALK*) fusion oncogene, and c-ROS oncogene 1 (*ROS1*) fusions. In addition, current studies revealed the existence of inherited cancer syndrome, which results in lung cancer susceptibility; among those inherited genes, mutations in *P53* and the epidermal growth factor receptor (*EGFR*) were best documented [17,18,19,20,21].

In 2001 imatinib was the first epidermal growth factor receptor tyrosine kinase inhibitor (EGFR-TKI) approved by the U.S. FDA; it proved to be a pivotal drug in the oncology area and beyond [22]. Two decades later, there are 71 small molecular kinase inhibitors approved by the U.S. FDA [23]. For advanced NSCLCs harboring activating EGFR mutations, an EGFR-TKI provides a superior survival benefit compared with platinum-based chemotherapy [24,25]. Osimertinib, another potent EGFR-TKI, was shown to be highly active in patients who had resistance to previous EGFR inhibitors [26]. Suresh et al., demonstrated the benefit of Osimertinib in progression-free survival as a first-line treatment of EGFR mutation advanced NSCLC [27]. In the northern European study, with the available EGFR TKI drug (ATC code) were gefinitib (L01XE02), erlotinib (L01XE03), afatinib (L01XE13), and the median OS was slightly prolonged by approximal one month in the observational period 2010 to 2015 [28]. These results reflected cumulative evidence in EGFR-TKI treatments. Taiwan’s National Insurance approved erlotinib in 2013. Afatinib has been approved in 2014. To our disappointment, the National Health Insurance program approved the use of Osimertinib in 2020, which is out of our study period.

The National Health Insurance program approved gefitinib in 2006. A previous Taiwan study indicated significant changes in lung cancer mortality 3 years after the launch of gefitinib, which is consistent with our study [29]. Our study showed that increased overall survival in adenocarcinoma, especially in the early stages, resulted in a total increased overall survival among lung cancer patients. Although EGFR-TKIs had great therapeutic efficacy in early-stage adenocarcinoma, little improvement in late stages (stage III, stage IV) was observed in this study. One reason is that resistance to first-generation TKIs usually occurred 9–13 months after treatment started [30]. Another reason is the diversity of EGFR mutations.

Adenosquamous cell carcinoma (LUADSC), which accounts for 0.4–4% of all lung cancers, is a rare and aggressive form of non-small cell lung carcinoma. Literature reported that EGFR mutations were seen in 13.1–33.3% of tumor specimens. Moreover, the objective response rate was 26.5%, and the disease control rate was 65.3% with EGFR-TKI treatments such as gefitinib or erlotinib [31,32]. This study showed increased 3-year OS among LUADSC patients, indicating the therapeutic efficacy of EGFR-TKIs.

However, Fang et al., reported that EFGR mutations in Chinese squamous cell carcinoma patients are extremely uncommon [33]. Thus, there is no significant advance in OS regarding squamous cell carcinoma. In addition, large cell carcinoma (LCC) rarely expressed EGFR mutations. A Chinese study reviewed 24 surgically resected LCC patients. Instead of EGFR mutations, P53 account for half of the somatic mutations [34]. Pulmonary small cell carcinomas have unique features and also seldom express EGFR mutations. EGFR mutations occur in 2 out of 11 small cell carcinoma patients. Therefore, current EGFR-TKIs were not suitable treatments [35].

Our study has several strengths. First, the large number of patients provides certain power to identify minor differences between subgroups. Second, target therapy is the pivotal point in treating different histopathologies of lung cancer. In this study, we found the efficacy of targeted therapy in improving the overall survival of adenocarcinoma patients, which contributed to the increase in overall lung cancer survival from 2010 to 2016. Our literature provides a new aspect of using different target therapies for different cell types. Third, this study could result in new National Health Insurance approval of different target regimens for different histological types of lung cancer.

Our study also has some limitations. The first one is the retrospective nature of the study. The second limitation is the inability to capture neoadjuvant treatment or immunotherapies. The chemotherapy regimens, target therapy, surgical skills, and radiotherapy protocols varied among different hospitals in Taiwan. These heterogeneous treatment protocols may also have confounded the study results. Third, the inability to retrieve the smoking history and genetic background of primary tumors.

## 5. Conclusions

To summarize, more frequent use of thoracic computed tomography, resulted in more small size (<2 cm) tumors being diagnosed. And surgery is the main treatment option for early-stage lung cancer. The percentage of patients receiving surgical intervention rose from 16.4% (2002–2008) to 28.2% (2010–2016); this contributed to the five-year survival rate increasing from 57.19% to 69.93%. Early detection, early surgical intervention, and cooperation with the administration of EGFR-TKIs, which is 0.1% in 2010 and 29.8% in 2016, respectively. Such intervention had a tremendous improvement in adenocarcinoma patients around the world. we can see a future of increased overall survival from different cell types of lung cancer.

## Figures and Tables

**Figure 1 jcm-11-05503-f001:**
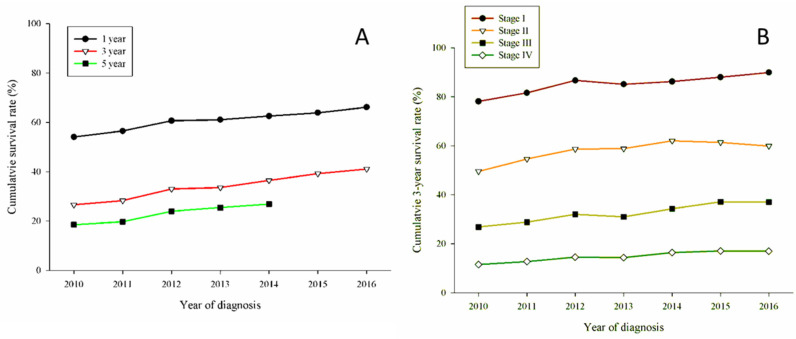
(**A**) Cumulative survival rates for 71,334 patients with lung cancer; *(p* < 0.01); A trend of gradually increased survival rates in 3 years and 5 years, (**B**) Cumulative 3-year survival rate stratified by stage of all patients. Stage I had the best 3-year cumulative survival rates above 80% in 2011, and Stage IV remain the least during the observation period.

**Figure 2 jcm-11-05503-f002:**
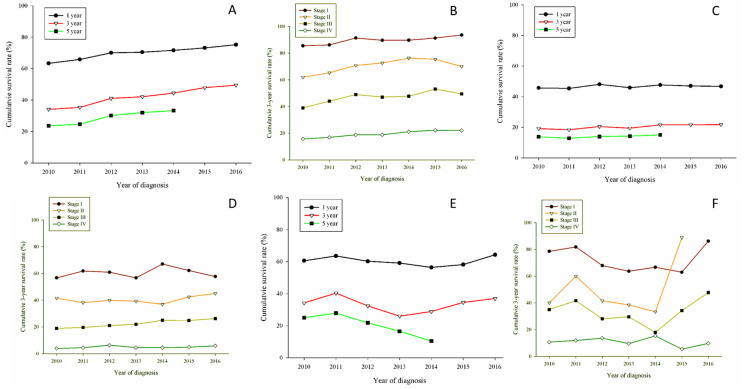
Kaplan-Meier overall survival by histopathology. (**A**) Cumulative survival rates by years in total of 46,783 adenocarcinoma patients, survival rate gradually increased in 1-year,3-year, and 5-year. (**B**) 3-year cumulative survival rates by year stratified by stage of adenocarcinoma patients. Stage I has the best survival rate. Stage II gradually increased from 2010 to 2015 and declined in 2016. Stage III had a fluctuating pattern between the observational period. Stage IV had a cumulative survival rate of no more than 20%. (**C**) Cumulative survival rates by year for 11,005 squamous cell carcinoma patients. Compared with adenocarcinoma, no significant increase in cumulative survival rates. (**D**) 3-year cumulative survival rates stratified by squamous cell carcinoma patients. Only in 2014, the cumulative survival rate reaches 60% in stage I. Compared with adenocarcinoma, there’s still a need for more advanced treatment options. (**E**) Cumulative survival rates by year for 869 adenosquamous cell carcinoma patients. Stage I patients had a survival rate above 60% as adenocarcinoma, but the 3-year and 5-year cumulative survival significantly decreased. (**F**) 3-year cumulative survival rates stratified by stage of adenosquamous cell carcinoma patients. Cumulative survival rates started to increase in all stages. Started in the year 2014 in stage II patients, or 2015 in stage I patients.

**Figure 3 jcm-11-05503-f003:**
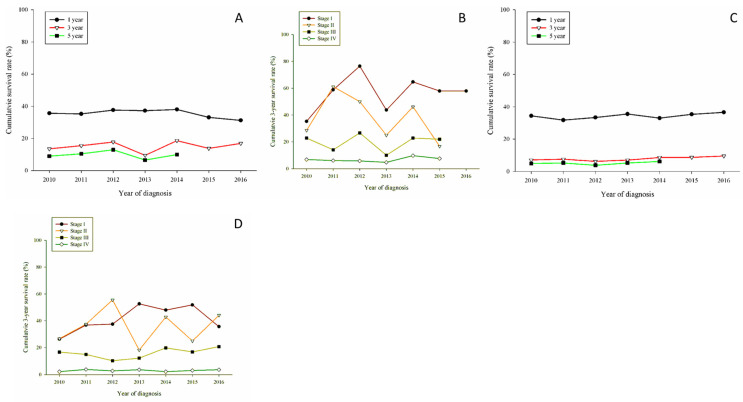
(**A**) Cumulative survival rates for 1774 large cell carcinoma patients, obviously poor prognosis compared with adenocarcinoma patients. (**B**) 3-year cumulative survival rates stratified by stage of large cell carcinoma patients. A fluctuating pattern was seen in stage I, stage II, and stage III. (**C**) Cumulative survival rates for 5565 small cell carcinoma patients. Tremendous poor prognosis compared with adenocarcinoma and squamous cell carcinoma. (**D**) 3-year cumulative survival rates stratified by stage of small cell carcinoma patients. Stage IV had a cumulative survival rate of less than 10%. Slightly increased in the observation period in stage I patients. A fluctuating pattern was observed in stage II patients.

**Table 1 jcm-11-05503-t001:** Basic data of lung cancer patients in Taiwan from 2010 to 2016.

Features	2010	2011	2012	2013	2014	2015	2016
Patients (N)	9260	9105	9711	10,019	10,621	11,053	11,565
Age (mean ± SD)	68.03 ± 12.79	67.41 ± 12.61	67.09 ± 12.71	66.90 ± 12.62	66.71 ± 12.59	66.95 ± 12.40	66.61 ± 12.26
Gender							
Male	5789 (62.5%)	5603 (61.5%)	5726 (59.0%)	5949 (59.4%)	6108 (57.5%)	6325 (57.2%)	6434 (55.6%)
Female	3471 (37.5%)	3502 (38.5%)	3985 (41.1%)	4070 (40.6%)	4513 (42.5%)	4728 (42.8%)	5131 (44.4%)
CCI score							
≤2	1390 (15.1%)	1396 (15.3%)	1654 (17.0%)	1776 (17.7%)	1925 (18.1%)	2073 (18.8%)	2759 (23.9%)
3–4	2446 (26.4%)	2372 (26.1%)	2552 (26.3%)	2677 (26.7%)	2720 (25.6%)	2933 (26.5%)	3213 (27.8%)
5–8	2050 (22.1%)	2023 (22.2%)	2106 (21.7%)	2139 (21.4%)	2303 (21.7%)	2305 (20.9%)	2374 (20.5%)
>8	3374 (36.4%)	3314 (36.4%)	3399 (35.0%)	3427 (34.2%)	3673 (34.6%)	3742 (33.9%)	3219 (27.8%)
Location							
Right	5218 (56.4%)	5233 (57.5%)	5590 (57.6%)	5681 (56.7%)	6141 (57.8%)	6305 (57.0%)	6603 (57.1%)
Left	3922 (42.3%)	3790 (41.6%)	4016 (41.3%)	4235 (42.3%)	4348 (40.9%)	4639 (42.0%)	4845 (41.9%)
Bilateral	34 (0.4%)	25 (0.3%)	32 (0.3%)	58 (0.6%)	75 (0.7%)	43 (0.4%)	49 (0.4%)
Missing	86 (0.9%)	57 (0.6%)	73 (0.8%)	45 (0.5%)	57 (0.5%)	66 (0.6%)	68 (0.6%)
Cell type							
LUAD	5421 (58.5%)	5607 (61.6%)	6323 (65.1%)	6655 (66.4%)	7192 (67.7%)	7528 (68.1%)	8060 (69.7%)
LUSC	1646 (17.8%)	1595 (17.5%)	1540 (15.9%)	1544 (15.4%)	1581 (14.9%)	1580 (14.3%)	1519 (13.1%)
LUADSC	76 (0.82%)	104 (1.14%)	133 (1.37%)	127 (1.27%)	142 (1.34%)	136 (1.23%)	151 (1.31%)
LCC	311 (3.36%)	258 (2.83%)	231 (2.38%)	212 (2.12%)	263 (2.48%)	275 (2.49%)	224 (1.94%)
SCC	787 (8.50%)	793 (8.71%)	751 (7.73%)	794 (7.92%)	794 (7.48%)	802 (7.26%)	844 (7.30%)
Others	981 (10.59%)	719 (7.90%)	696 (7.17%)	644 (6.43%)	606 (5.71%)	691 (6.25%)	734 (6.35%)
Tumor size							
<2 cm	807 (8.7%)	878 (9.6%)	1138 (11.7%)	1257 (12.6%)	1514 (14.3%)	1746 (15.8%)	2131 (18.4%)
2–3 cm	1272 (13.7%)	1294 (14.2%)	1489 (15.3%)	1544 (15.4%)	1705 (16.1%)	1804 (16.3%)	1771 (15.3%)
3–4 cm	1361 (14.7%)	1315 (14.4%)	1433 (14.8%)	1530 (15.3%)	1579 (14.9%)	1555 (14.1%)	1646 (14.2%)
4–5 cm	1069 (11.5%)	1125 (12.4%)	1208 (12.4%)	1177 (11.7%)	1245 (11.7%)	1281 (11.6%)	1320 (11.4%)
5–7 cm	1527 (16.5%)	1528 (16.8%)	1475 (15.2%)	1615 (16.1%)	1673 (15.8%)	1761 (15.9%)	1712 (14.8%)
7–9 cm	778 (8.40%)	883 (9.70%)	865 (8.91%)	831 (8.29%)	908 (8.55%)	938 (8.49%)	961 (8.31%)
≥9 cm	485 (5.24%)	508 (5.58%)	518 (5.33%)	515 (5.14%)	524 (4.93%)	566 (5.12%)	592 (5.12%)
Missing	1961 (21.2%)	1574 (17.3%)	1585 (16.3%)	1550 (15.5%)	1473 (13.9%)	1402 (12.7%)	1432 (12.4%)
Clinical stage							
IA	800 (8.6%)	837 (9.2%)	1120 (11.5%)	1216 (12.1%)	1487 (14.0%)	1767 (16.0%)	1981 (17.1%)
IB	451 (4.9%)	486 (5.3%)	545 (5.6%)	579 (5.8%)	643 (6.1%)	669 (6.1%)	714 (6.2%)
IIA	261 (2.8%)	231 (2.5%)	218 (2.2%)	242 (2.4%)	233 (2.2%)	241 (2.2%)	276 (2.4%)
IIB	207 (2.2%)	179 (2.0%)	208 (2.1%)	188 (1.9%)	225 (2.1%)	231 (2.1%)	216 (1.9%)
IIIA	826 (8.9%)	734 (8.1%)	760 (7.8%)	751 (7.5%)	754 (7.1%)	754 (6.8%)	730 (6.3%)
IIIB	926 (10.0%)	857 (9.4%)	917 (9.4%)	876 (8.7%)	899 (8.5%)	933 (8.4%)	908 (7.9%)
IV	5546 (59.9%)	5635 (61.9%)	5767 (59.4%)	5873 (58.6%)	6147 (57.9%)	6184 (56.0%)	6306 (54.5%)
Unknown	243 (2.6%)	146 (1.6%)	176 (1.8%)	294 (2.9%)	233 (2.2%)	274 (2.5%)	434 (3.8%)
Treatment							
Any	7778 (84.0%)	8396 (92.2%)	8986 (92.5%)	9340 (93.2%)	9891 (93.1%)	10,282 (93.0%)	10,763 (93.1%)
C/T	6163 (66.6%)	5099 (56.0%)	4958 (51.1%)	4886 (48.8%)	4901 (46.1%)	4791 (43.4%)	4907 (42.4%)
Surgery	1945 (21.0%)	2107 (23.1%)	2580 (26.8%)	2824 (28.2%)	3174 (29.9%)	3527 (31.9%)	3974 (34.4%)
RT	2612 (28.2%)	2831 (31.1%)	2879 (29.7%)	2897 (28.9%)	3029 (28.5%)	3018 (27.3%)	3046 (26.3%)
Target	8 (0.1%)	2552 (28.0%)	2987 (30.8%)	3164 (31.6%)	3389 (31.9%)	3364 (30.4%)	3450 (29.8%)

CCI: Charlson Comorbidity Index; LUAD: Lung adenocarcinoma; LUADSC: Lung adenosquamous cell carcinoma; LUSC: Lung squamous cell carcinoma; LCC: large cell carcinoma; SCC: small cell carcinoma; C/T: chemotherapy; RT: radiotherapy.

**Table 2 jcm-11-05503-t002:** Overall survival rate of lung cancer patients in Taiwan from 2010 to 2016.

	2010	2011	2012	2013	2014	2015	2016
All patients							
1-year OS	54.07 (53.05–55.09)	56.49 (55.47–57.51)	60.63 (59.66–61.60)	61.09 (60.14–62.04)	62.56 (61.64–63.48)	63.91 (63.01–64.81)	66.14 (65.28–67.00)
3-year OS	26.57 (25.67–27.47)	28.27 (27.34–29.20)	32.98 (32.05–33.91)	33.56 (32.63–34.49)	36.48 (35.56–37.40)	39.25 (38.34–40.16)	41.12 (39.85–42.39)
5-year OS	18.5 3(17.74–19.32)	19.78 (18.96–20.60)	24.01 (23.16–24.86)	25.45 (24.60–26.30)	26.86 (25.61–28.11)		
Median survival (months)	13.75 (13.30–14.20)	15.27 (14.73–15.82)	17.92 (17.33–18.52)	18.31 (17.71–18.92)	19.85 (19.14–20.56)	21.81 (21.01–22.61)	24.68 (23.67–25.69)
1-year OS							
Clinical stage I	92.57 (91.12–94.02)	94.0 3(92.75–95.31)	95.92 (94.97–96.87)	94.54 (93.49–95.59)	96.10 (95.28–96.92)	96.39 (95.65–97.13)	96.7 3(96.06–97.40)
Clinical stage II	74.36 (70.40–78.32)	79.51 (75.61–83.41)	82.86 (79.27–86.45)	83.95 (80.48–87.42)	84.28 (80.95–87.61)	85.59 (82.41–88.77)	84.15 (80.92–87.38)
Clinical stage III	58.7 3(56.42–61.04)	61.28 (58.89–63.67)	65.30 (63.03–67.57)	61.52 (59.15–63.89)	66.06 (63.79–68.33)	65.68 (63.41–67.95)	66.00 (63.71–68.29)
Clinical stage IV	41.35 (40.05–42.65)	44.10 (42.80–45.40)	46.59 (45.30–47.88)	47.64 (46.36–48.92)	47.47 (46.22–48.72)	47.85 (46.61–49.09)	49.68 (48.45–50.91)
3-year OS							
Clinical stage I	78.18 (75.89–80.47)	81.6 3(79.55–83.71)	86.7 3(85.10–88.36)	85.1 3(83.48–86.78)	86.24 (84.78–87.70)	88.05 (86.76–89.34)	89.9 3(88.29–91.57)
Clinical stage II	49.57 (45.04–54.10)	54.6 3(49.81–59.45)	58.69 (54.01–63.37)	58.84 (54.19–63.49)	62.01 (57.56–66.46)	61.44 (57.05–65.83)	59.92 (52.14–67.70)
Clinical stage III	26.8 3(24.75–28.91)	28.79 (26.56–31.02)	32.08 (29.85–34.31)	30.98 (28.73–33.23)	34.30 (32.01–36.59)	37.11 (34.80–39.42)	37.01 (33.85–40.17)
Clinical stage IV	11.54 (10.70–12.38)	12.76 (11.89–13.63)	14.53 (13.62–15.44)	14.39 (13.49–15.29)	16.40 (15.47–17.33)	17.09 (16.15–18.03)	17.01 (15.49–18.53)
5-year OS							
Clinical stage I	68.35 (65.76–70.94)	71.28 (68.85–73.71)	76.76 (74.72–78.80)	76.55 (74.59–78.51)	77.17 (74.94–79.40)		
Clinical stage II	39.53 (35.10–43.96)	41.95 (37.17–46.73)	43.90 (39.20–48.60)	46.05 (41.35–50.75)	42.08 (28.09–56.07)		
Clinical stage III	16.95 (15.19–18.71)	19.74 (17.78–21.70)	20.21 (18.29–22.13)	21.39 (19.39–23.39)	25.13 (22.82–27.44)		
Clinical stage IV	4.53 (3.98–5.08)	5.16 (4.58–5.74)	7.02 (6.36–7.68)	7.0 3(6.38–7.68)	7.10 (5.82–8.38)		

Note: overall survival rates are given as percentages.

## Data Availability

Restrictions apply to the availability of these data. Data was obtained from Taiwan Cancer Registry and are available from Bing-Yen Wang with the permission of Taiwan Cancer Registry.

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
