# Peer review of "Survival of Lung Cancer Patients by Histopathology in Taiwan from 2010 to 2016: A Nationwide Study"

_jcm, 2022, doi:10.3390/jcm11195503_

Round 1

Reviewer 1 Report

1. Authors should clearly explain why they have focused on a limited period between 2010-2016. Why is it no longer? Already we are in 2022 and in my opinion indicating the overview of the last decade 2010-2020 would be much more interesting and valuable. The threshold at 2020 would be also better due to the fact of COVID-19 breakthrough that changed the lung cancer statics due to the various issues with availability to screening, diagnosis and treatment.

2. The study focused on a Taiwan cohort that seems to be more regional interest. Authors should clearly highlighted why the proposed topic is interesting for worldwide audience and discuss their results with outcomes from other populations.

3. The abbreviations used in the text should be in accordance with TCGA nomenclature that become the most common used worldwide. Ex Lung adenocarcinoma = LUAD; lung squamous cell carcinoma = LUSC. Moreover, all genes name should be written in italics.

4. Table 1 should be supplemented by smoking history and genetic background of primary tumors. Showing statistic for these factors would be very interesting. It would be also informative how the trends of used therapies changed in each cancer type within the presented period.

5. The figures do not present the KM curves, in my opinion they plot only the survival trends calculated by other method – could author double check the methodology used at the plots? Anyway, these data should be implemented by relevant statistic calculations. Moreover, the figures captions should be more detailed elaborating deeply the presented data. It is also not clear what is the difference between Figure 2 and 3.

6. It would be extremely interesting to indicated more multivariate correlations between subtypes of lung cancer and clinical/demographic factors that are already collected.

7. The discussion is not confirmed by strong statistic data in the paper. There should be also more comparisons between trends changes in US and European cohorts.

8. The description of LDCT in the discussion part seem to be not located in the proper place, also providing description about application of LDCT between 2019-2022 is out of the scope (time period) of the paper. Also adding the immunotherapy part into discussion is not proper while statistic for this treatment regimen was not included.

9. The conclusions do not arise clearly form the data presented in the paper.

Author Response

Responses to the reviewers’ comments on our manuscript (Submission ID: jcm-1910918)

I appreciate the reviewers’ comments. Our point-by-point responses to the comments are listed below (the line and page numbers are shown for convenience, and the revised material is indicated in red text in the marked version of revised manuscript).

 Reviewer #1:

  1. Authors should clearly explain why they have focused on a limited period between 2010-2016. Why is it no longer? Already we are in 2022 and in my opinion indicating the overview of the last decade 2010-2020 would be much more interesting and valuable. The threshold at 2020 would be also better due to the fact of COVID-19 breakthrough that changed the lung cancer statics due to the various issues with availability to screening, diagnosis and treatment.

Response:  Difference edition of TNM classification

We thank this reviewer’s comment.  We choose period 2010 -2016 by the reason of different edition of the American Joint Committee on Cancer (AJCC). Year 2010-to 2016 is the seventh edition of the TNM classification of malignant tumors system. And the Eighth edition of the TNM classification was enacted in 2017.

Because when comparing the survival of cancer, it is best to use the same version of the TNM staging, if different versions of the TNM staging are mixed, it will cause confusion, because each version of the staging, the definition of the stage is not the same

Reference.

  1. Seventh edition: Goldstraw, P., et al., (2007). The IASLC Lung Cancer Staging Project: proposals for the revision of the TNM stage groupings in the forthcoming (seventh) edition of the TNM Classification of malignant tumours. Journal of thoracic oncology : official publication of the International Association for the Study of Lung Cancer, 2(8), 706–714.
  2. Eighth edition: Goldstraw, P., et al., International Association for the Study of Lung Cancer Staging and Prognostic Factors Committee, Advisory Boards, and Participating Institutions, & International Association for the Study of Lung Cancer Staging and Prognostic Factors Committee Advisory Boards and Participating Institutions (2016). The IASLC Lung Cancer Staging Project: Proposals for Revision of the TNM Stage Groupings in the Forthcoming (Eighth) Edition of the TNM Classification for Lung Cancer. Journal of thoracic oncology: official publication of the International Association for the Study of Lung Cancer, 11(1), 39–51.
  3. The study focused on a Taiwan cohort that seems to be more regional interest. Authors should clearly highlighted why the proposed topic is interesting for worldwide audience and discuss their results with outcomes from other populations.

Response:  Page 12 line 34-48

We thank this reviewer’s comment. We have the added following sentence in the “discussion” paragraph.

Sheng Hu et al disclose the same trend in OS of adenocarcinoma patient using the US Surveillance, Epidemiology, and End Results (SEER) database (1998-2018), 27.8% of the US population were cover by SEER data-base. They discovered an 41.72% increase in median survival time of all lung cancer patients. Adenocarcino-ma patients had a 5% increase in median survival comparing with 3 % increase in squamous patients.[12] [12] OS of American population statistics in adenocarcinoma and squamous cell carcinoma was consistent with our study, A 2.71% increase in SEER database. We revealed no significant differences in overall survival curve of squamous patients. (Fig 2C).

Likewise, five-year relative survival ratio in adenocarcinoma increased both in men and women, 10% in men, 8% in women respectively form year 1996-2016. Contrary to the finding in U.S. and Asia, squamous cell carcinoma account mostly in Estonian population.[13]In northern Europe, adenocarcinoma accounted for half of all lung cancers, the study also seen an improvement in relative survival both in adenocarcinoma and squamous cell carcinoma. In contrary, squamous cell carcinoma had better prognosis than adenocarcinoma.[14] The survival differs among different region around the world. Asian and United States share similar results.

3.The abbreviations used in the text should be in accordance with TCGA nomenclature that become the most common used worldwide. Ex Lung adenocarcinoma = LUAD; lung squamous cell carcinoma = LUSC. Moreover, all genes name should be written in italics.

Response:  Whole article

We thank this reviewer’s comment.

We had changed the abbreviations in accordance with TCGA nomenclature.

  1. Table 1 should be supplemented by smoking history and genetic background of primary tumors. Showing statistic for these factors would be very interesting. It would be also informative how the trends of used therapies changed in each cancer type within the presented period

Response: Page 13, Line137-138 Limitations

We thank this reviewer’s comment. We have carefully read the suggested article and added following sentence in every “Discussion” paragraph.

Third, the inability to retrieve the smoking history and genetic background of primary tumors

  1. The figures do not present the KM curves; in my opinion they plot only the survival trends calculated by other method – could author double check the methodology used at the plots? Anyway, these data should be implemented by relevant statistic calculations. Moreover, the figures captions should be more detailed elaborating deeply the presented data. It is also not clear what is the difference between Figure 2 and 3.

Response: Explanation, and add figure captions,

We thank this reviewer’s comment. We have carefully read the suggested article and added following sentence in every “figure captions” paragraph.

Our Kaplan Meier survival curve was not presented to the reviewers, though survival rate generated by Kaplan Meier was presented and further distinguished differences from previous survival ratings via x square test

Figure 1. (A) Cumulative survival rates for 71,334 patients with lung cancer;(P<0.01); A trend of graudlly increased survival rates in 3 year and 5 year (B) Cumulative 3-year survival rate stratified by stage of all patients. Stage I had the best 3-year cumulative survival rates above 80% in the year of 2011, Stage IV remain the least as the observation period.

Figure 2

  • Cumulative survival rates by years in total46,783 adenocarcinoma patients, survival rate gradually increased in 1-year,3-year, and 5-year.
  • 3-year cumulative survival rates by year stratified by stage of adenocarcinoma patients. Stage I has the best survival rate. Stage II gradually increased from 2010 to 2015 and decline in 2016. Stage III had a fluctuate pattern between the observational period. Stage IV had a cumulative survival rate no more than 20%.
  • Cumulative survival rates by year for 11,005 squamous cell carcinoma patients. Compared with adenocarcinoma, no significant increase in cumulative survival rates.
  • 3-year cumulative survival rates stratified by of squamous cell carcinoma patients. Only in 2014, the cumulative survival rate reaches 60% in stage I. Compared with adenocarcinoma, there’s still need more advanced treatment options.
  • Cumulative survival rates by year for 869 adenosquamous cell carcinoma patients. Stage I patients had a survival rate above 60% as adenocarcinoma, but the 3-year and 5-year cumulative survival significant decreased.

(F) 3-year cumulative survival rates stratified by stage of adenosquamous cell carcinoma patients. Cumulative survival rates started to increase in all stage. Started from the year 2014 in stage II patients, 2015 in stage I patients.

Figure 3

  • Cumulative survival rates for 1774 large cell carcinoma patients, obvious poor prognosis compared with adenocarcinoma patients.
  • 3-year cumulative survival rates stratified by stage of large cell carcinoma patients. A fluctuated pattern was seen in stage I, stage II and stage III.
  • Cumulative survival rates for 5565 small cell carcinoma patients. Tremendous poor prognosis compared with adenocarcinoma, squamous cell carcinoma.
  • 3-year cumulative survival rates stratified by stage of small cell carcinoma patients. Stage IV had the cumulative survival rate less than 10 %. Slightly increased in the observation period in stage I patients. A fluctuated pattern was observed in stage II patients.
  1. It would be extremely interesting to indicated more multivariate correlations between subtypes of lung cancer and clinical/demographic factors that are already collected.

Response: Explanation

We thank this reviewer’s comment.

Because there are so many subtypes of lung cancer, multivariate analysis had its difficulty. We believe that squamous cell carcinoma and adenocarcinoma have different clinical/demographic factors, and their prognostic factors and treatment methods are also different. If only the two most common subtypes were analyzed, we have published such an article. “The comparison between adenocarcinoma and squamous cell carcinoma in lung cancer patients” in 2020.

Reference

  1. Wang BY, Huang JY, Chen HC, Lin CH, Lin SH, Hung WH, Cheng YF. The comparison between adenocarcinoma and squamous cell carcinoma in lung cancer patients. J Cancer Res Clin Oncol. 2020 Jan;146(1):43-52
  2. The discussion is not confirmed by strong statistic data in the paper. There should be also more comparisons between trends changes in US and European cohorts.

Response: Page 12, Line 34-48

We thank this reviewer’s comment. We have the added following sentence in the “discussion” paragraph.

Sheng Hu et al disclose the same trend in OS of adenocarcinoma patient using the US Surveillance, Epidemiology, and End Results (SEER) database (1998-2018), 27.8% of the US population were cover by SEER database. They discovered an 41.72% increase in median survival time of all lung cancer patients. Adenocarcino-ma patients had a 5% increase in median survival comparing with 3 % increase in squamous patients.[12] OS of American population statistics in adenocarcinoma and squamous cell carcinoma was consistent with our study, A 2.71% increase in SEER database. We revealed no significant differences in overall survival curve of squamous patients. (Fig 2C).

Likewise, five-year relative survival ratio in adenocarcinoma increased both in men and women, 10% in men, 8% in women respectively form year 1996-2016. Contrary to the finding in U.S. and Asia, squamous cell carcinoma account mostly in Estonian population.[13] In northern Europe, adenocarcinoma accounted for half of all lung cancers, the study also seen an improvement in relative survival both in adenocarcinoma and squamous cell carcinoma. In contrary, squamous cell carcinoma had better prognosis than adenocarcinoma.[14] The survival differs among different region around the world. Asian and United States share similar results.

  1. The description of LDCT in the discussion part seem to be not located in the proper place, also providing description about application of LDCT between 2019-2022 is out of the scope (time period) of the paper. Also adding the immunotherapy part into discussion is not proper while statistic for this treatment regimen was not included.

Response: Page 12, Line 62-65. Page 12, Line 118-123

We thank this reviewer’s comment. We have the deleted following sentence in the “discussion” paragraph.

Page 11

In 2019 in Taiwan, the Taichung City government started a program using LDCT for detecting early lung cancer in police and firefighters aged 40 and above. Likewise, in 2022 the Taiwan government implemented a program that offered a free LDCT exam to people who have a family history of lung cancer and have smoked at least 30 pack-years.

Page 13

Furthermore, inhibitors of the programmed death-1 (PD-1) immune checkpoint re-ceptor are currently remodeling the landscape of NSCLC treatment. The U.S. FDA has ap-proved nivolumab (Opdivo, Bristol-Myers Squibb) and pembrolizumab (Keytruda, Merck Sharp and Dohme) for advanced NSCLC in patients previously treated with plati-num-based chemotherapy.[397, 4038] A Taiwan literature showed that among adenocar-cinoma patients, those with no programmed death-ligand-1 (PD-L1) expression had better survival than those with PD-L1 expression or EGFR mutations.[4139]

  1. The conclusions do not arise clearly form the data presented in the paper.

Response: Page 13, Line 126-133

We thank this reviewer’s comment. We have the added following sentence in the “conclusions” paragraph.

  To summarize, more frequent use of thoracic computed tomography, resulted in  more small size (<2cm) tumor were diagnosed. And surgery is the main treatment op-tion for early-stage lung cancer. The percentage of patients receiving surgical intervention rose from 16.4% (2002-2008) to 28.2% (2010-2016); this contributed to the five-year survival rate increasing from 57.19% to 69.93%. Early detection, early surgical intervention, and cooperate with the administration of EGFR-TKIs, which is 0.1% in 2010 and 29.8% in 2016, respectively. Such intervention had a tremendous improvement in adenocarcinoma patient around the world. we can see a future of increased overall survival from several types of lung cancer.

Reviewer 2 Report

In light of the significant changes over the past 10 years in lung cancer diagnosis and treatment it is interesting to see good national population register based studies such as this. The intro/methods and results section is fine, easy to follow and my only comment is that legend F for fig 2 is missing and that legend D for fig 3 is also missing incl the text apparently.

My main problem with this manuscripts is the structure of the discussion.  First the authors talk about screening but clearly screening have not affected the presented data so this part seems irrelevant. So what other reasons can give the observed change in size and stage??Then follows a section on histological type and molecular changes. Rather than talking about the history of TKI, more discussion of EGFR would be appropriate such as osimertinib use in the period. Also if the authors want to talk about immune checkpoint inhibitors it really should be in the context of their study: how could the ICI's have affected the survival in the period???  I simply dont understand the conclusion, no systematic screening was done in the period and the conclusion should be related to the results.

Author Response

Responses to the reviewers’ comments on our manuscript (Submission ID: jcm-1910918)

I appreciate the reviewers’ comments. Our point-by-point responses to the comments are listed below (the line and page numbers are shown for convenience, and the revised material is indicated in red text in the marked version of revised manuscript).

 Reviewer #2:

my only comment is that legend F for fig 2 is missing and that legend D for fig 3 is also missing incl the text apparently.

My main problem with this manuscripts is the structure of the discussion.  First the authors talk about screening but clearly screening have not affected the presented data so this part seems irrelevant. So what other reasons can give the observed change in size and stage??Then follows a section on histological type and molecular changes. Rather than talking about the history of TKI, more discussion of EGFR would be appropriate such as osimertinib use in the period. Also if the authors want to talk about immune checkpoint inhibitors it really should be in the context of their study: how could the ICI's have affected the survival in the period???  I simply dont understand the conclusion, no systematic screening was done in the period and the conclusion should be related to the results.

Response:  Page 12, Line 50-59

We thank this reviewer’s comment. We have added following sentence in the “discussion” paragraph.

Advanced lung cancer predominates in the whole worlds. In an eastern European study, a slightly increased localized lung cancer was discovered in 2010 to 2016. The proportion of localized cases of adenocarcinoma increased with approximately 2% in a eastern European study. However, there has probably been a shift toward smaller tumors within this rather well-studied category, more frequent use of chest computed tomography (CT) for other indications has certain contribution. and resulting incidental detection of early tumors. From 2010 to 2016, the proportion of T1 tumors among localized cases had a 20% increased.[13] China researchers also confirmed with the im-provement in diagnostic techniques such as radiologic imaging, and CT, more patients are being detected in early stage of NSCLC during 1998-2005.[15] We could estimated more early stage of NSCLC were detected with the advancement of technology.

Page 12, 77-79

Likewise, we could see the tumor size below 2 cm increased from 8.7% in 2010 to 14.3% in 2016, respectively. The stage 1A patients increased from 8.6% in 2010 to 14% in 2016.

We thank this reviewer’s comment. We have deleted following sentence in the “discussion” paragraph

Response: Page 12 Line 49-60

Lung cancer was the leading cause of cancer death in the United States, but the mor-tality had been declining in recent years.[15] Early screening for lung cancer with low-dose computed tomography (LDCT) has its role. LDCT was first described in the 1980s. A U.S. study demonstrated that with a low-radiation dose, which is one-tenth of the normal chest CT radiation dose, a high-quality diagnostic image of a lung could be ob-tained.[16, 17]

In a worldwide review, Pinsky indicated that early-implemented LDCT screening, with necessary quality controls and monitoring, may be an option to reduce lung cancer mortality in certain high-risk populations.[18] U.S. studies revealed that using LDCT for screening in high-risk smokers, who had a history of cigarette smoking for at least 30 pack-years, reduces lung cancer mortality by 20%.[19, 20] The same effect could be seen in Asia.[21]

Response: Page 13, Line 104-112

Osimertinib, another potent EGFR-TKI, shown highly active in patient had resistance to previous EGFR inhibitors.[26] Suresh et al demonstrated the benefit of Osimertinib in progression-free sur-vival  as a first-line treatment of EGFR mutation advanced NSCLC.[27] In northern Europe study, with the available EGFR TKI drug (ATC code) were gefinitib (L01XE02),erlotinib (L01XE03), afatinib(L01XE13), the median OS slightly prolonged by approximal one month in the observa-tional period 2010 to 2015.[28] These results reflected a cumulative evidence in EGFR-TKI treatments. Taiwan’s National Insurance approved erlotinib in 2013.Afatinib been approved in 2014. To our disappointing, the National Health Insurance program approved use of Osimertinib in 2020, which is out of our study period.

We thank this reviewer’s comment. We have the added following sentence in the “conclusions” paragraph.

Response: Page 14, Line 152-160

To summarize, more frequent use of thoracic computed tomography, resulted in  more small size (<2cm) tumor were diagnosed. And surgery is the main treatment op-tion for early-stage lung cancer. The percentage of patients receiving surgical intervention rose from 16.4% (2002-2008) to 28.2% (2010-2016); this contributed to the five-year survival rate increasing from 57.19% to 69.93%. Early detection, early surgical intervention, and cooperate with the administration of EGFR-TKIs, which is 0.1% in 2010 and 29.8% in 2016, respectively. Such intervention had a tremendous improvement in adenocarcinoma patient around the world. we can see a future of increased overall survival from several types of lung cancer.

Round 2

Reviewer 1 Report

Authors have replied to all of my remarks in a satisfactory way, however I would like to suggest one minor change that could make the text more uniform.

1. In all places where adenocarcinoma or squamous cell/squamous is used it would be better to provide LUAD and LUSC abbreviation according to TCGA nomenclature that is widely used by the community.

Author Response

Responses to the reviewers’ comments on our manuscript (Submission ID: jcm-1910918)

I appreciate the reviewers’ comments. Our point-by-point responses to the comments are listed below (the line and page numbers are shown for convenience, and the revised material is indicated in red text in the marked version of revised manuscript).

  1. In all places where adenocarcinoma or squamous cell/squamous is used it would be better to provide LUAD and LUSC abbreviation according to TCGA nomenclature that is widely used by the community

Response: Page 3, Line 106

We have corrected the wrong abbreviation. 

And change AdSqcc to LUADSC to make the text more uniform.

 The prevalence of adenocarcinoma increased by 11 percentage points, and there were gradual decreases in the prevalence of squamous cell carcinoma (LUSC), large cell carcinoma (LCC), and other cell types of lung cancer.
